# Associations of First-Trimester Screening Markers and Hematological Indices with Placenta Accreta Spectrum in Pregnancies Complicated by Placenta Previa

**DOI:** 10.3390/biomedicines13092082

**Published:** 2025-08-27

**Authors:** Volkan Karatasli, Ahkam Goksel Kanmaz, Alaattin Karabulut, Abdurrahman Hamdi Inan

**Affiliations:** 1Department of Obstetrics and Gynecology, Balikesir Ataturk City Hospital, Balikesir 10100, Turkey; 2Department of Obstetrics and Gynecology, Izmir City Hospital, Izmir 35540, Turkey; drgokselkanmaz@gmail.com; 3Department of Obstetrics and Gynecology, Tepecik Education and Research Hospital, Izmir 35100, Turkey; alaattin_karabulut@hotmail.com (A.K.); ahamdiinan@gmail.com (A.H.I.)

**Keywords:** first-trimester screening, free β-hCG, MPV, PAPP-A, placenta accreta spectrum, placenta previa, prenatal diagnosis

## Abstract

**Background**: Placenta accreta spectrum (PAS) is a serious pregnancy complication associated with significant hemorrhaging and elevated maternal morbidity. Timely prenatal diagnosis is critical for reducing the risk of adverse outcomes. In this study, we aimed to investigate the association between PAS and first-trimester maternal serum screening markers, as well as selected hematological and inflammatory indices, in pregnancies complicated by placenta previa (PP). **Methods**: A retrospective study was conducted at a tertiary care center. Pregnant women with singleton pregnancies who had been diagnosed with PP and undergone first-trimester aneuploidy screening and delivered at the same institution were included. The participants were divided into two groups: those diagnosed with PAS (including placenta accreta, increta, and percreta) and those with PP without placental invasion. Data on maternal demographics, the first-trimester serum levels of pregnancy-associated plasma protein-A (PAPP-A), and free β-human chorionic gonadotropin (β-hCG), as well as pre-delivery complete blood count parameters, were collected. Associations between these markers and abnormal placental implantation were analyzed. **Results**: In total, 181 participants were included in this study, corresponding to 15 cases of PAS and 166 cases of non-invasive PP. The women in the PAS group were significantly younger than those in the non-invasive-PP group (25.3 ± 5.1 vs. 30.0 ± 6.3 years, *p* < 0.001). The serum levels of PAPP-A and free β-hCG were significantly higher in the PAS cases (*p* < 0.05). The mean platelet volume (MPV) was significantly lower inF the PAS group (*p* < 0.05). We did not observe any significant differences in other hematological parameters, including hemoglobin concentration, white blood cell count, neutrophil and lymphocyte counts, platelet count, red cell distribution width, and inflammatory ratios such as the neutrophil-to-lymphocyte and platelet-to-lymphocyte ratios. **Conclusions**: Elevated first-trimester levels of PAPP-A and β-hCG, along with a reduced MPV, may serve as early indicators of PAS in pregnancies complicated by PP. These biomarkers may assist in early risk stratification and help inform perinatal management strategies.

## 1. Introduction

Placenta previa (PP), defined as the partial or complete coverage of the internal cervical os by placental tissue, is a significant risk factor for maternal and fetal morbidity, particularly in the third trimester [1]. PP occurs in approximately 0.3–0.5% of pregnancies [2]. One of the most serious complications associated with PP is placenta accreta spectrum (PAS), characterized by abnormal placental adherence or invasion into the myometrium, ranging in severity from accreta to percreta [2]. The development of PAS is strongly linked to a history of uterine surgery, particularly cesarean delivery, and its incidence has risen markedly in parallel with the global increase in cesarean section rates [3,4]. It is currently estimated to occur in approximately 1 in 588 deliveries [5]. PAS is associated with severe obstetric complications, including massive peripartum hemorrhaging, disseminated intravascular coagulation, multiorgan failure, hysterectomy, and elevated maternal and neonatal mortality rates [6]. Early identification of PAS is critical for optimizing outcomes, as planned deliveries at tertiary care centers with experienced multidisciplinary teams have been shown to significantly reduce morbidity and improve survival [1]. Despite advancements in imaging, many PAS cases continue to be undiagnosed until delivery, often resulting in life-threatening hemorrhaging [1,7].

Recent studies have proposed that the pathogenesis of PAS is a complex process involving abnormal trophoblastic invasion, impaired decidualization, excessive neovascularization, and chronic inflammation at the maternal–fetal interface [8,9]. It has also been suggested that a persistently inflamed endometrial environment may promote aberrant trophoblastic behavior, contributing to the development of PAS [10]. In this context, inflammatory and angiogenic markers detectable in maternal blood may offer early insights into abnormal placentation before imaging or clinical signs emerge [9].

First-trimester maternal serum biomarkers—such as pregnancy-associated plasma protein-A (PAPP-A) and free β-human chorionic gonadotropin (β-hCG), which are routinely used in aneuploidy screening—reflect trophoblastic function and placental development [9]. They have been associated with adverse obstetric outcomes [11]. However, their specific predictive value for PAS remains uncertain and has yet to be clearly established [6,12]. Similarly, systemic inflammatory markers derived from complete blood count parameters—such as the neutrophil-to-lymphocyte ratio (NLR), platelet-to-lymphocyte ratio (PLR), mean platelet volume (MPV), and red cell distribution width (RDW)—have been investigated regarding their potential roles as indicators of systemic inflammation and obstetric complications [13]. However, their utility for predicting abnormal placental invasion has not been firmly established, as few studies have evaluated the relationship between these markers and the severity of placental invasion [14,15].

Few studies have specifically addressed high-risk pregnancies complicated by PP, where differentiating invasive forms of PAS from non-invasive forms is vital [5]. Most research has assessed either first-trimester biochemical markers or third-trimester hematological indices in isolation, without integrating both early and late markers [9].

We hypothesize that abnormal placentation and the associated inflammatory milieu in PAS may be reflected by alterations in first-trimester serum markers (PAPP-A and β-hCG) and third-trimester hematological indices (especially MPV). Therefore, in pregnancies complicated by PP, these markers might serve as early indicators of PAS, enabling risk stratification before delivery.

In this study, we aim to evaluate the predictive value of selected first- and third-trimester maternal blood markers in distinguishing PAS from non-invasive PP, seeking to inform cost-effective early screening strategies, which are particularly valuable in low-resource settings.

## 2. Materials and Methods

### 2.1. Study Design

In this retrospective study, we evaluated deliveries at a tertiary care center between January 2013 and January 2017. Ethical approval was obtained from the local ethics committee (approval no.: 2020/2-3, dated 12 February 2020), and the study was conducted in accordance with the Declaration of Helsinki. Informed consent was obtained from all the participants.

### 2.2. Participants

This study included pregnant women with singleton pregnancies diagnosed with PP in hospital records who had undergone first-trimester serum aneuploidy screening and delivered at the same center. The exclusion criteria were as follows: multiple pregnancies; stillbirths; fetal anomalies; chromosomal disorders; neural tube defects; chronic conditions, such as cardiac disease, diabetes mellitus, hypertension, and autoimmune diseases; and a history of drug use. The participants were also excluded if they had incomplete clinical data, were smokers, had acute or chronic infections, or experienced pregnancy complications such as preeclampsia, gestational diabetes mellitus, preterm premature rupture of membranes, or intrauterine growth restriction (IUGR). Additionally, women who had a history of PP, undergone previous cervical or uterine surgery, or conceived via assisted reproductive technologies were excluded.

Only live births with a confirmed diagnosis of PP and complete clinical data were included. All the patients underwent cesarean delivery. The participants were divided into two groups: those diagnosed with PAS (including placenta accreta, increta, or percreta) and those with PP without evidence of placental invasion. Demographic and clinical data were collected, including maternal age, body mass index, gravidity, parity, medical and surgical history, gestational age at delivery, neonatal birth weight, and admission to a neonatal intensive care unit. In addition, operative records, blood transfusion requirements, first-trimester screening test results, and pre-delivery complete blood count parameters were analyzed.

Due to the retrospective nature of this study and the rarity of PAS, no formal a priori power analyses were conducted. Instead, all eligible and confirmed PAS cases during the study period were included. We acknowledge that the relatively small number of PAS cases may limit this study’s statistical power, especially for detecting small-to-moderate differences in secondary variables such as fetal sex distribution. The unequal sample sizes between groups reflect the true incidence of PAS in this population.

### 2.3. Diagnostic Criteria

PP was defined as partial or complete coverage of the internal cervical os by the placenta diagnosed via ultrasonography (transabdominal or transvaginal) or, when clinically indicated, magnetic resonance imaging and confirmed intraoperatively. The sonographic features suggestive of placenta accreta include the absence of the retroplacental clear zone, irregularity of the uteroplacental interface, and myometrial thinning of less than 1 mm [1,16]. The diagnosis of PAS was confirmed either histopathologically via surgical specimens or based on operative findings in cases managed conservatively. A definitive intraoperative diagnosis was established when the placental separation plane with respect to the uterine wall was indistinct, resulting in difficulty with manual removal, often accompanied by severe hemorrhaging [6].

### 2.4. Biomarker and Hematological Assessment

First-trimester serum levels of PAPP-A and free β-hCG were obtained through aneuploidy screening conducted between 11 + 0 and 13 + 6 weeks of gestation using an Immulite 2000 XPi analyzer (Siemens Healthcare Diagnostics, Los Angeles, CA, USA). Multiples of the median values were calculated using PRISCA 5.0 software, adjusted for maternal age, weight, gestational age, smoking status, and history of diabetes. Complete blood count results, collected within 24 h prior to delivery, included hemoglobin levels, hematocrit, white blood cell counts, neutrophil and lymphocyte counts, platelet counts, MPV, and RDW. NLR and PLR were calculated based on the aforementioned parameters. Associations between abnormal placental implantation and both first-trimester aneuploidy screening biomarkers and systemic inflammatory markers derived from complete blood count were analyzed.

### 2.5. Statistical Analysis

Statistical analyses were performed using SPSS version 21 (IBM Corp., Armonk, NY, USA). Continuous variables were presented as means ± standard deviations or as medians with ranges, depending on the distribution of the data. Group comparisons were conducted using the independent samples *t*-test for parametric data and the Mann–Whitney U test for non-parametric data. Categorical variables were analyzed using the chi-square test or Fisher’s exact test, as appropriate. Receiver operating characteristic (ROC) curve analysis was used to evaluate the diagnostic performance of hematological parameters in predicting placental invasion. A *p*-value < 0.05 was considered statistically significant.

## 3. Results

In total, 181 cases were included in this study, comprising 15 pregnancies affected by PAS and 166 complicated by non-invasive PP. Clinical parameters arranged according to the presence of placental invasion are summarized in Table 1. The women in the PAS group were significantly younger than those in the PP group (25.3 ± 5.1 vs. 30.0 ± 6.3 years, *p* < 0.001). The prevalence of a previous cesarean section was significantly higher in the PAS group (66.7%) relative to the PP group (7.2%) (*p* < 0.001). There were no significant differences between the groups in terms of gestational age at delivery, neonatal birth weight, or requirement for admission to a neonatal intensive care unit (*p* > 0.05).

Blood transfusion was significantly more common in the PAS group, with 93.3% of the patients requiring transfusions, while this figure was only 29.5% in the PP group (*p* < 0.001). Hysterectomy was performed in six cases in the PAS group. Hypogastric artery ligation was conducted in nine cases—three in the PAS group and six in the PP group. Additionally, urinary bladder injury repair was required in three cases in the PAS group.

Table 2 presents the results of the first-trimester screening tests. Serum PAPP-A levels were significantly higher in the PAS group relative to the PP group (*p* = 0.013). Likewise, serum β-hCG levels were significantly elevated in the PAS group (*p* = 0.024).

Table 3 presents the hematologic parameters. MPV was significantly lower in the PAS group relative to the PP group (*p* = 0.037). No significant differences were observed between the groups in regard to other parameters, including hemoglobin, hematocrit, white blood cell count, neutrophil count, lymphocyte count, platelet count, RDW, NLR, and PLR. ROC curve analysis (Figure 1) was conducted to assess the predictive value of MPV for PAS. The area under the curve for MPV was 0.628, indicating limited diagnostic utility. MPV exhibited a negative correlation with PAS, suggesting that lower MPV values may be associated with an increased risk of abnormal placental invasion.

As shown in Table 4, the levels of the studied markers in PAS patients were compared with reference ranges reported for normal pregnancies in the literature.

Figure 2 shows boxplots of PAPP-A, β-hCG, and MPV, indicating higher PAPP-A and β-hCG levels and lower MPV values in the PAS cases compared to the PP controls.

## 4. Discussion

PAS is a serious pregnancy complication associated with life-threatening hemorrhaging and substantial maternal morbidity [5]. Early prenatal diagnosis is critical for reducing complications, as it allows for preoperative planning and timely referral to tertiary care centers equipped with multidisciplinary teams [1,5]. In this retrospective study, we investigated the potential utility of first-trimester maternal serum biomarkers (PAPP-A and β-hCG) and third-trimester hematologic-inflammatory indices (particularly MPV) in predicting PAS among pregnancies complicated by PP. Our findings suggest that elevated PAPP-A and β-hCG levels, as well as decreased MPV, are significantly associated with PAS and may serve as potential early indicators of abnormal placentation. These significant differences in both biochemical and hematological parameters between PAS cases and non-invasive PP cases underscore the value of employing laboratory-based risk stratification strategies alongside imaging. Our study is distinct in that it focuses solely on PP pregnancies—a high-risk subset—and incorporates both early and late pregnancy markers for a biphasic screening model.

PAPP-A, produced via the syncytiotrophoblast and modulating insulin-like growth factor activity essential for placental development, may reflect excessive trophoblastic invasion when elevated [9]. Elevated β-hCG levels may indicate increased trophoblastic mass or proliferative activity, both of which are characteristic of invasive placentation [9]. Additionally, the decreased MPV observed in PAS cases may result from chronic low-grade inflammation or platelet consumption due to abnormal vascular remodeling at the uteroplacental interface [14,20].

Previous studies investigating the diagnostic value of biomarkers potentially associated with PAS have examined parameters such as maternal serum alpha-fetoprotein, PAPP-A, and β-hCG [6,21]. However, none of these markers have demonstrated sufficient accuracy to support their routine use in clinical prediction [1]. In this study, PAPP-A and β-hCG levels were significantly higher in cases diagnosed with PAS. Although these markers are primarily utilized for aneuploidy screening, recent research has explored their utility in the early detection of placental pathologies. Several studies have reported elevated PAPP-A levels in PAS cases [17,22], although the findings have not been entirely consistent. For instance, Penzhoyan et al. [18] found no significant differences in PAPP-A levels between PAS and PP cases. A systematic review of 731 patients reported higher PAPP-A levels in placenta accreta cases compared to those in PP cases [6], supporting early-trimester biomarker assessment. Compared to previous studies, our research is unique in that it combines early and late markers and excludes confounders such as preeclampsia, infections, and IUGR [5,6]. Table 4 shows how our findings reflect as well as extend beyond standard reference ranges.

The predictive value of β-hCG in the diagnosis of PAS remains uncertain in the current literature. A systematic review and meta-analysis involving 733 patients reported that second-trimester β-hCG levels were significantly higher in PAS cases relative to those with PP [6]. However, the clinical relevance of first-trimester β-hCG levels is still inconclusive. Consistent with our findings, Buke et al. [22] also reported elevated β-hCG levels in PAS cases, whereas other studies found no significant differences in β-hCG concentrations between PAS and PP groups [18,23]. These inconsistencies underscore the need for larger, prospective studies.

Previous studies have explored the potential role of systemic inflammatory markers derived from complete blood count parameters—such as MPV, NLR, PLR, and RDW—in the diagnosis of PAS [15,24]. Yayla et al. and Keles et al. [14,25] reported higher MPV values in PAS cases, whereas in this study, consistent with the findings of Ersoy et al. [19], we observed lower MPV levels in cases with placental invasion. This discrepancy may be attributed to the influence of severe inflammation, which has been suggested to reduce MPV levels [20]. Although some studies have reported elevated NLR and PLR values in PAS patients [14,25], others—including this study—have not found any significant differences in these markers [19,26]. It has been proposed that lymphocyte depletion may contribute to abnormal placental invasion [14]. Additionally, a recent study suggested that elevated first-trimester RDW levels may aid in predicting PAS [24]. However, in this study, preoperative RDW values did not differ significantly between groups, a finding consistent with that reported by Keles et al. [14], though Yayla et al. reported lower RDW levels in PAS cases [25]. These inconsistencies may be explained by differences in study populations, the timing of measurements, or methodological variability. Unlike the NLR or PLR, which are non-specific and variable during pregnancy, MPV may represent a more sensitive indicator of the subtle pathophysiological changes in PAS [24]. Given that PAS is associated with various inflammatory changes during its pathogenesis [14], further large-scale prospective studies are needed.

Though elevations in PAPP-A and β-hCG levels are not exclusive to PAS and may also be observed in conditions such as aneuploidy, preeclampsia, or gestational trophoblastic disease, their interpretation in the specific context of PP may improve diagnostic specificity [6]. A multimodal approach combining biochemical markers with ultrasound and maternal history enhances diagnostic accuracy, which is especially crucial in low-resource settings with limited imagining capabilities [5].

Previous cesarean delivery was significantly more frequent among PAS cases in this study, reaffirming its role as the most important risk factor for abnormal placental attachment [4]. Although advanced maternal age (>35 years) is traditionally considered a major risk factor for PAS, the PAS group in this study was significantly younger than the PP group [27]. This finding may reflect increasing cesarean delivery rates among younger women and shifts in population demographics. Similar results were reported by Oztas et al. [2,28], suggesting that factors such as multiparity and surgical history may have a greater influence on PAS development than chronological age in certain populations. Increased blood transfusions and surgical complications such as hysterectomy and bladder injuries further highlight the surgical risks of PAS, justifying prenatal suspicion and multidisciplinary planning [1,27,29,30].

This study has several limitations. First, the sample size for PAS cases was relatively small (n = 15), reflecting the rarity of the condition. Although our analysis suggested significant associations, the limited number of cases may have reduced the statistical power and restricted the generalizability of the findings. Second, a control group of normal pregnancies was not included, as our primary aim was to differentiate invasive from non-invasive PP. Future studies incorporating healthy controls could provide valuable insight into the behavior of these markers across the full spectrum of placentation. Third, as this is a retrospective single-center study, selection bias cannot be excluded, and the results may not be directly applicable to other populations or healthcare settings. Finally, we did not evaluate additional angiogenic or proteomic markers, which may have further enhanced the predictive accuracy of our model. Despite these limitations, this study provides valuable clinical insights and highlights the potential role of biochemical and hematological markers in assessing the risk of PAS, supporting the need for further prospective, multicenter research.

## 5. Conclusions

This study highlights the potential utility of first-trimester maternal serum markers and hematological parameters in identifying pregnant women at increased risk for PAS in the context of PP. Elevated levels of PAPP-A and β-hCG were significantly associated with PAS, indicating their value as early indicators of abnormal placental invasion. Additionally, the observation of lower MPV in PAS cases points to a possible association between platelet indices and placental pathology. Although these findings support further investigation into biochemical and hematological screening tools, their limited diagnostic accuracy underscores the need for larger, prospective studies. Integrating such markers into existing risk assessment protocols may enhance early recognition and facilitate management planning, particularly in resource-limited settings where imaging expertise or access to specialized care is constrained.

Larger, prospective multicenter studies including normal pregnancy controls are required to validate these findings. The integration of biochemical markers with sonographic features, maternal history, and novel proteomic or angiogenic biomarkers could enhance early diagnostic accuracy for PAS. Additionally, exploring machine learning models incorporating these variables may offer promising decision-support tools in prenatal care. Future research should focus on validating these associations and developing multiparameter screening models to improve the prenatal prediction of PAS.

## Figures and Tables

**Figure 1 biomedicines-13-02082-f001:**
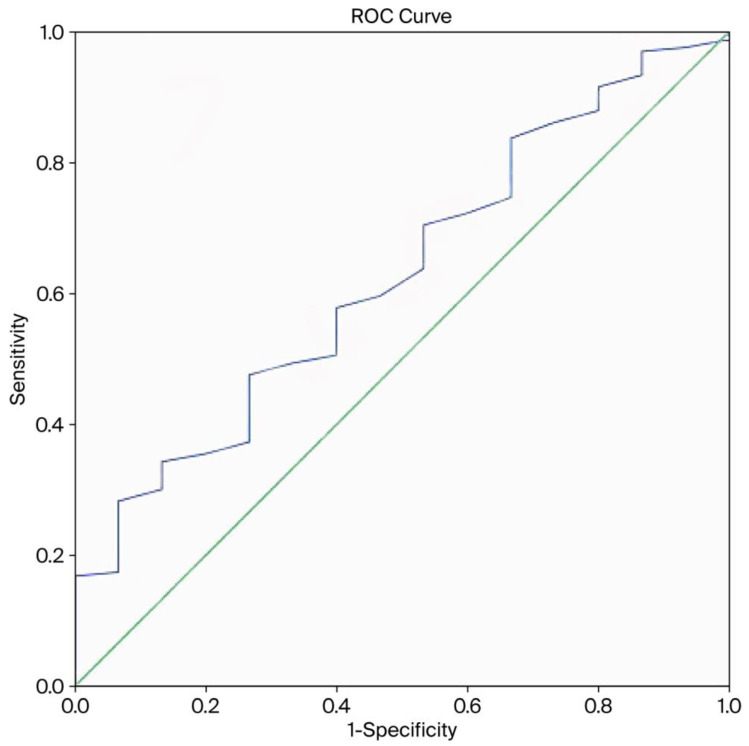
Receiver operating characteristic (ROC) curve for the predictive ability of mean platelet volume (MPV) for placenta accreta spectrum.

**Figure 2 biomedicines-13-02082-f002:**
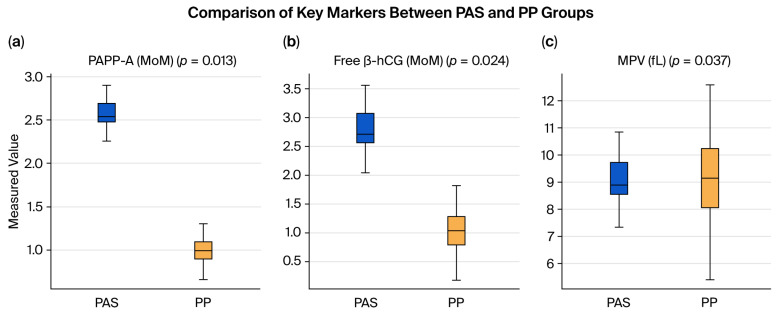
Boxplots of biochemical and hematological markers in the placenta accreta spectrum (PAS) and placenta previa (PP) groups. The graphs depict differences in (**a**) PAPP-A levels, (**b**) free β-hCG (MoM) levels, and (**c**) mean platelet volume (MPV). Statistical comparisons were performed using the Mann–Whitney U test.

**Table 1 biomedicines-13-02082-t001:** Demographic and clinical variables pertaining to the study participants.

	Placenta Accreta Spectrum (n = 15)	Non-Adherent Placenta Previa (n = 166)	*p* Value
Age (years), mean ± SD	25.3 ± 5.1	30.0 ± 6.3	**<0.001**
BMI (kg/m^2^), mean ± SD	23.6 ± 1.7	24.0 ± 1.9	0.358
Gravida, mean ± SD	3.0 ± 0.4	3.1 ± 0.5	0.254
Parity, n (%)			
Nulliparity	2 (13.3%)	47 (28.3%)	0.173
Multiparity	13 (86.7%)	119 (72.7%)	
Fetal sex, n (%)			
Male	9 (60%)	83 (50%)	0.458
Female	6 (40%)	83 (50%)	
Prior cesarean section, n (%)			
Yes	10 (66.7%)	12 (7.2%)	**<0.001**
No	5 (33.3%)	154 (92.8%)	
Gestational age at birth (weeks), mean ± SD	36.3 ± 1.3	35.6 ± 3.0	0.475
Birth weight (g), mean ± SD	2869 ± 640.9	2656 ± 701.0	0.237
Admission to neonatal intensive care unit, n (%)			
Yes	1 (6.7%)	30 (18.1%)	0.232
No	14 (93.3%)	136 (81.9%)	
Blood transfusion, n (%)			
Yes	14 (93.3%)	49 (29.5%)	**<0.001**
No	1 (6.7%)	117 (70.5%)	
Surgical complications, n (%)			
Hysterectomy	6 (40.0%)	0 (0%)	**<0.001**
Hypogastric artery ligation	3 (20.0%)	6 (3.6%)	**0.005**
Urinary bladder injury	3 (20.0%)	0 (0%)	**<0.001**

SD = standard deviation; BMI = body mass index. Bold values are statistically significant; *p* < 0.05 was considered indicative of statistical significance.

**Table 2 biomedicines-13-02082-t002:** First-trimester screening test results according to placental invasion status.

	Placenta Accreta Spectrum (n = 15)	Non-Adherent Placenta Previa (n = 166)	*p* Value
PAPP-A (MoM), mean ± SD	2.45 ± 0.2	0.99 ± 0.13	**0.013**
Free β-hCG (MoM), mean ± SD	2.5 ± 0.47	1.03 ± 0.33	**0.024**

SD = standard deviation; PAPP-A = pregnancy associated plasma protein-A; β-hCG = beta human chorionic gonadotropin; MoM = multiples of the median. Bold values are statistically significant; *p* < 0.05 was considered indicative of statistical significance.

**Table 3 biomedicines-13-02082-t003:** Preoperative complete blood count parameters according to placental invasion status.

	Placenta Accreta Spectrum (n = 15)	Non-Adherent Placenta Previa (n = 166)	*p* Value
WBC count (10^3^/μL), median (range)	11.6 (7.1–13.5)	11.2 (6.1–28.5)	0.945
Neutrophil count (10^3^/μL), median (range)	8.5 (5.4–10.9)	8.2 (2.5–25.3)	0.951
Lymphocyte count (10^3^/μL), median (range)	2.1 (1.2–2.7)	2 (0.4–6.6)	0.633
Hemoglobin (g/dL), mean ± SD	10.7 ± 1.25	11.1 ± 1.23	0.284
Hematocrit (%), mean ± SD	32.9 ± 3.88	33.7 ± 3.52	0.433
Platelet count (10^3^/μL), median (range)	228 (127–327)	220 (81–515)	0.582
MPV (fL), mean ± SD	8.4 ± 1.09	9.1 ± 1.45	**0.037**
RDW (%), median (range)	15.9 (12.7–22.8)	14.2 (12.0–23.7)	0.099
NLR, median (range)	3.56 (2.67–9.08)	4.13 (0.38–24.0)	0.740
PLR, median (range)	108.0 (60.4–230	111.4 (33.3–420)	0.975

SD = standard deviation; WBC = white blood cell; MPV = mean platelet volume; RDW = red blood cell distribution width; NLR = neutrophil-to-lymphocyte ratio; PLR = platelet-to-lymphocyte ratio. Bold values are statistically significant; *p* < 0.05 was considered indicative of statistical significance.

**Table 4 biomedicines-13-02082-t004:** Comparison of our findings with reference values in normal pregnancies.

	Placenta Accreta Spectrum (Our Study)	Normal Pregnancy (Literature) *	Reference
PAPP-A (MoM), mean ± SD	2.45 ± 0.2	0.99 ± 0.13	Wang et al., 2021 [17]
Free β-hCG (MoM), mean ± SD	2.5 ± 0.47	1.08 ± 0.69 (0.5–2.0)	Penzhoyan et al., 2019 [18]
MPV (fL), mean ± SD	8.02 ± 0.94	10.54 ± 0.90 (7.4–13.1)	Ersoy et al., 2016 [19]

SD = standard deviation; PAPP-A = pregnancy associated plasma protein-A; β-hCG = beta human chorionic gonadotropin; MPV = mean platelet volume; MoM = multiples of the median. * The data are given as means ± SD (range).

## Data Availability

The data presented in this study are only available on request from the corresponding author for ethical reasons.

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
