# Peer review of "Associations of First-Trimester Screening Markers and Hematological Indices with Placenta Accreta Spectrum in Pregnancies Complicated by Placenta Previa"

_biomedicines, 2025, doi:10.3390/biomedicines13092082_

Round 1

Reviewer 1 Report

Comments and Suggestions for Authors

The manuscript entitled "Associations of First-Trimester Screening Markers and Hematological Indices with Placenta Accreta Spectrum in Pregnancies with Placenta Previa" describes how major biomarkers such as elevated PAPP-A and β-hCG, along with reduced MPV, may act as diagnosing factors for PAS. I have some suggestions/ concerns regarding the status. Please find the comments below:

  1. The authors need to state the inflammatory conditions and its possible mechanisms behind PAS in the introduction itself. It is necessary as the authors are looking at the inflammatory components as a possible biomarker. Hence, a proper hypothesis must be stated (refer https://pmc.ncbi.nlm.nih.gov/articles/PMC6051104/)
  2. There are several studies since 2018, indicating the aforementioned biomarkers as indicators of PAS. These studies include https://doi.org/10.1111/aogs.14918, https://doi.org/10.1186/s12884-023-05784-2, https://doi.org/10.3389/fmed.2022.860186 etc. What is the novelty of the present study in comparison with these studies?
  3. Authors need to include a table indicating the variation of the selected parameters in PAS with that of a normal pregnancy (based on the literature values and indicate as a range).
  4. The higher values of PAPP-A and β-hCG are indicative of various genetic and non-genetic diseases/ syndromes. How the condition of PAS can be differentiated based on the values of these.
  5. Figure 1 quality is low and not visible properly
  6. Authors need to strengthen the discussion section, as the selected markers are general in nature. A specific possible reason is to be indicated why the selected biomarkers indicate PAS in the first trimester.

Author Response

Comments 1: “The authors need to state the inflammatory conditions and its possible mechanisms behind PAS in the introduction itself. It is necessary as the authors are looking at the inflammatory components as a possible biomarker. Hence, a proper hypothesis must be stated. (refer https://pmc.ncbi.nlm.nih.gov/articles/PMC6051104/).”

Response 1: Thank you for pointing this out. We agree with this comment. Therefore, we have revised the Introduction section to include a more detailed background on the proposed inflammatory mechanisms contributing to PAS development. Based on the references provided (Bartels et al., 2018 and Verdugo et al., 2024), we now discuss the role of aberrant trophoblastic invasiondeficient decidualization, and chronic inflammation in the uterine environment, which may lead to pathologic neovascularization and disruption of the normal uteroplacental barrier.

Furthermore, we have clearly stated our hypothesis in the revised manuscript:

“We hypothesize that abnormal placentation and the associated inflammatory milieu in PAS may be reflected by alterations in first-trimester serum markers (PAPP-A and β-hCG) as well as in third-trimester hematological indices (especially MPV). Therefore, in pregnancies complicated by PP, these markers might serve as early indicators of PAS, enabling risk stratification before delivery.”

(See revised Introduction, page 2, lines 60–66 and 83–87)

Comments 2: “There are several studies since 2018, indicating the aforementioned biomarkers as indicators of PAS. These studies include https://doi.org/10.1111/aogs.14918, https://doi.org/10.1186/s12884-023-05784-2, https://doi.org/10.3389/fmed.2022.860186 etc. What is the novelty of the present study in comparison with these studies?”

Response 2: Thank you for pointing this out. We appreciate the reviewer highlighting these important publications. We have now revised the Discussion section and clarified the novelty of our work. While prior studies have explored individual biomarkers in the general obstetric population, our study is unique in that it focuses specifically on pregnancies complicated by placenta previa (PP)—a high-risk subgroup for PAS. Furthermore, unlike some previous studies that assessed second-trimester or peripartum markers, we analyzed first-trimester PAPP-A and β-hCG in combination with third-trimester hematological indices, offering a biphasic biomarker assessment model. Additionally, the study design includes a strict exclusion of confounding variables (e.g., preeclampsia, IUGR, infections), which enhances the specificity of associations found with PAS.

We have now emphasized these aspects in the revised Introduction and Discussion.

(See Introduction, page 2, lines 79–82 and Discussion, page 8, lines 226–228).

Comments 3: “Authors need to include a table indicating the variation of the selected parameters in PAS with that of a normal pregnancy (based on the literature values and indicate as a range).”

Response 3: Thank you for pointing this out. We agree that this addition enhances the interpretability of our findings. We have now included Table 4 in the Results section, which compares our study’s findings on PAPP-A, β-hCG, and MPV values in PAS cases to literature-reported values in normal pregnancies. Ranges for normal pregnancy are based on recent studies.

(See newly added Table 4, page 7)

Comments 4: “The higher values of PAPP-A and β-hCG are indicative of various genetic and non-genetic diseases/syndromes. How the condition of PAS can be differentiated based on the values of these?”

Response 4: Thank you for pointing this out. We agree with this comment. It is true that elevated PAPP-A and β-hCG levels can be associated with a range of pregnancy complications. However, our study specifically evaluates these biomarkers in the context of confirmed placenta previa, a known risk factor for PAS. We have clarified in the Discussion that the predictive value of these biomarkers for PAS is not based on absolute levels, but rather on contextual interpretation, in conjunction with clinical and imaging findings. We also excluded the participants if they had fetal anomalies, chromosomal disorders, neural tube defects, or experienced pregnancy complications such as preeclampsia, gestational diabetes mellitus, preterm premature rupture of membranes, or intrauterine growth restriction. Moreover, we emphasize that PAPP-A and β-hCG alone are not diagnostic; instead, they may serve as adjunctive markers in a multifactorial risk assessment strategy, particularly useful in resource-limited settings. This point has been expanded in the revised Discussion.

(see Discussion, page 8, lines 247–250 and 278–283).

Comments 5: “Figure 1 quality is low and not visible properly.”

Response 5: Thank you for pointing this out. We appreciate the feedback. Figure 1 has been replaced with a high-resolution version. The revised figure is compliant with the journal’s graphical standards.

(See revised Figure 1, page 6)

Comments 6: “Authors need to strengthen the discussion section, as the selected markers are general in nature. A specific possible reason is to be indicated why the selected biomarkers indicate PAS in the first trimester.”

Response 6: Thank you for pointing this out. We agree with this comment. Therefore, we have expanded the Discussion to include specific pathophysiological mechanisms that could explain the elevation of PAPP-A and β-hCG in PAS cases in the first trimester. In particular: PAPP-A is produced by syncytiotrophoblasts and is involved in insulin-like growth factor (IGF) regulation, which may be upregulated in cases of excessive trophoblastic invasion, as seen in PAS. β-hCG, another syncytiotrophoblast-derived hormone, may reflect increased trophoblastic mass or activity, potentially associated with invasive placental phenotypes. Additionally, we discuss that lower MPV levels may reflect systemic inflammatory changes or platelet consumption due to ongoing low-grade inflammation in PAS.

These interpretations are supported with relevant references and are included in the revised Discussion.

(see Discussion, page 8, lines 229–235 and 273-275).

Reviewer 2 Report

Comments and Suggestions for Authors

An interesting experimental study in which the authors studied the relationship between PAS and screening markers of maternal blood serum in the first trimester of pregnancy, as well as some hematological and inflammatory parameters in pregnancy complicated by placental presentation (PP). The work was performed on a sample of 181 pregnant women - 15 cases of RAS and 166 cases of noninvasive PP. The authors concluded that elevated levels of PAPP-A and β-hCG in the first trimester, along with a decrease in MPV, may serve as early indicators of PAS in pregnancy complicated by PP. These biomarkers may be of great practical importance as they will allow for early risk stratification and optimize the perinatal management strategy for women at risk. There are the following questions and suggestions about this work: 1. In the introduction of the work, data on the frequency of pregnancy complications under study should be given - PAS, PP. 2. Is the sample analyzed in the work - 15 cases of RAS - representative? How did the authors calculate the required sample size for the study? This should be described in the materials and methods. What kind of power does the sample being studied provide? Did a sample of 15 pregnant women lead to false negative results when differences of even 20% (for example, in the distribution of fetal sex) were not statistically significant? 3. Is there any data on the relationship between age and the concentration of biochemical and hematological markers studied in the work? And if this data is available, is it necessary to enter age as a covariate in the calculations in order to exclude its influence on the results obtained (since the two samples studied in the work differ significantly in average age)? 4. what is the average duration of pregnancy in the studied samples? This should be indicated in table 1.

Author Response

Comments 1: “In the introduction of the work, data on the frequency of pregnancy complications under study should be given - PAS, PP.”

Response 1: Thank you for pointing this out. In the revised version of the manuscript, we have now included epidemiological data on the prevalence of placenta previa (PP) and placenta accreta spectrum (PAS) disorders in the Introduction section. Specifically, we state that:

“Placenta previa occurs in approximately 0.3–0.5% of pregnancies.” and “PAS is currently estimated to occur in approximately 1 in 588 deliveries.”

(See Introduction page 2, lines 46 and 51–52)

Comments 2: “Is the sample analyzed in the work – 15 cases of PAS – representative? How did the authors calculate the required sample size for the study? This should be described in the materials and methods. What kind of power does the sample being studied provide? Did a sample of 15 pregnant women lead to false negative results when differences of even 20% (for example, in the distribution of fetal sex) were not statistically significant?”

Response 2: Thank you for pointing this out. We acknowledge the reviewer’s concern regarding sample size and statistical power. As noted, PAS is a relatively rare condition, and our study included all eligible cases from a single tertiary care center over a four-year period (2013–2017), resulting in 15 confirmed PAS cases among 181 total patients with placenta previa. To address this concern, we have now added a statement in the Materials and Methods section clarifying this point:

“Due to the retrospective nature of the study and the rarity of PAS, a formal a priori power analysis was not performed. Instead, all eligible and confirmed PAS cases during the study period were included. We acknowledge that the relatively small number of PAS cases may limit the statistical power, especially for detecting small-to-moderate differences in secondary variables such as fetal sex distribution.”

We further elaborated on the limitations of statistical power due to the sample size in the revised Discussion section.

(See Materials and Methods page 3, line 119–124 and Discussion page 9, lines 295–305)

Comments 3: “Is there any data on the relationship between age and the concentration of biochemical and hematological markers studied in the work? And if this data is available, is it necessary to enter age as a covariate in the calculations in order to exclude its influence on the results obtained (since the two samples studied in the work differ significantly in average age)?”

Response 3: Thank you for pointing this out. Indeed, maternal age differed significantly between the PAS and PP groups. To explore its potential confounding effect, we first examined correlations between maternal age and the studied markers; no statistically significant associations were observed for PAPP-A, β-hCG, or MPV. Subsequently, we performed sensitivity multivariable logistic regression analyses with PAS as the dependent variable, including maternal age as a covariate. In these adjusted models, the associations between PAS and the serum/hematological markers were attenuated and did not remain statistically significant. However, the reliability of these results is limited by the retrospective design, the small number of PAS cases, and the non-homogeneous distribution of the study groups, which likely reduced statistical power. For this reason, we did not consider it appropriate to include this particular analysis. This point has now been acknowledged in the revised manuscript, and the limitation has been explicitly stated in the Discussion section.

“This study has several limitations. First, the sample size of PAS cases was relatively small (n = 15), reflecting the rarity of the condition. Although the analysis suggested significant associations, the limited numbers may reduce statistical power and restrict the generalizability of the findings.”

(See Discussion page 9, lines 295–304)

Comments 4: “What is the average duration of pregnancy in the studied samples? This should be indicated in Table 1.”

Response 4: Thank you for pointing this out. The mean gestational age at delivery is presented in Table 1, being 36.3 ± 1.3 weeks for the PAS group and 35.6 ± 3.0 weeks for the PP group. The difference between the groups was not statistically significant (p = 0.475). This point is addressed in the Results section of the manuscript.

(See Results page 4, lines 168)

Reviewer 3 Report

Comments and Suggestions for Authors
  1. In the Keywords, authors can abbreviate pregnancy-associated plasma protein-A (PAPP-A). The given keywords are very general and suggested to give more specific keywords (Biomarker, haematological indices – are very general)
  2. Some latest literature work could have been included to state the epidemiological background of PAS and PP. There is a lack of citations in a few parts of the manuscript.
  3. How were the hematological-inflammatory indices related to the abnormal placentation? Explain.
  4. The hypothesis could have been written more effectively to explain the concept of this research.
  5. Presenting a graphical abstract in the manuscript will be easy for the readers to understand the hypothesis.
  6. Consider breaking the methodology parts into subheadings (e.g., Study Design, Participants, Diagnostic Criteria, Biomarker Assessment, Statistical Analysis).
  7. Why there is no control pregnant women are not included in this study?
  8. Is there any specific reason for large differences in the sample size for PAS, and PP.
  9. How the sample size was determined (based on power analysis or effect size). It should be explained in the sample collection section.
  10. There is a scope to include a graph or histogram in this manuscript.
  11. Among various hematological parameters, there is a significant change observed only in the MPV parameter? Give justification.
  12. Give a brief note on future research directions and how they could address the present study’s limitations in the conclusion.
  13. The ethical committee approval was given in the year 2020, but the article is communicated in 2025, is there any specific reason for this delay?

Author Response

Point-by-point response to Comments and Suggestions for Authors

Comments 1: “In the Keywords, authors can abbreviate pregnancy-associated plasma protein-A (PAPP-A). The given keywords are very general and suggested to give more specific keywords (Biomarker, haematological indices – are very general).”

Response 1: Thank you for pointing this out. We agree with this comment. In the revised manuscript, we have updated the Keywords section to include more specific and relevant terms. The abbreviation “PAPP-A” has been used, and general terms such as “biomarker” and “hematological indices” have been replaced or supplemented with more focused keywords such as:

placenta accreta spectrum, placenta previa, PAPP-A, free β-hCG, MPV, first-trimester screening, prenatal diagnosis.

(See Keywords page 1, lines 40–41)

Comments 2: “Some latest literature work could have been included to state the epidemiological background of PAS and PP. There is a lack of citations in a few parts of the manuscript.”

Response 2: Thank you for pointing this out. We appreciate this observation. In the Introduction and Discussion sections, we have now incorporated recent epidemiological data and references from the past five years to strengthen the scientific background of PAS and PP. Additional citations have been added in previously under-referenced areas to provide better support for the study rationale and interpretation.

“Placenta previa occurs in approximately 0.3–0.5% of pregnancies.” and “PAS is currently estimated to occur in approximately 1 in 588 deliveries.”

(See Introduction page 2, lines 46 and 51–52)

Comments 3: “How were the hematological-inflammatory indices related to the abnormal placentation? Explain.”

Response 3: Thank you for pointing this out. We have added a clearer explanation in the Introduction and Discussion sections. Hematological-inflammatory indices, such as MPV, reflect systemic inflammatory responses and vascular remodeling. Abnormal placentation in PAS may involve excessive trophoblastic invasion and impaired vascular remodeling, which could alter platelet activation and inflammation markers, including MPV. This biological plausibility supports our findings of reduced MPV in PAS cases.

(See Introduction page 2, lines 60–66 and Discussion page 8, lines 229–235 and 273–275)

Comments 4: “The hypothesis could have been written more effectively to explain the concept of this research.”

Response 4: Thank you for pointing this out. We agree with this comment and have revised the hypothesis statement in the Introduction section to better articulate the concept of the study:

“We hypothesize that abnormal placentation and the associated inflammatory milieu in PAS may be reflected by alterations in first-trimester serum markers (PAPP-A and β-hCG) as well as in third-trimester hematological indices (especially MPV). Therefore, in pregnancies complicated by PP, these markers might serve as early indicators of PAS, enabling risk stratification before delivery.

(See Introduction page 2, lines 83–87)

Comments 5: “Presenting a graphical abstract in the manuscript will be easy for the readers to understand the hypothesis.”

Response 5: Thank you for pointing this out. We appreciate this suggestion. A graphical abstract has now been created and included in the revised manuscript to visually summarize the study design, hypothesis, and main findings. We believe this will enhance the reader’s comprehension of our work.

Comments 6: “Consider breaking the methodology parts into subheadings (e.g., Study Design, Participants, Diagnostic Criteria, Biomarker Assessment, Statistical Analysis).”

Response 6: Thank you for pointing this out. We have revised the Materials and Methodssection to include clear subheadings as follows: Study Design, Participants, Diagnostic Criteria, Biomarker and Hematological Assessment and Statistical Analysis. This restructuring improves clarity and readability.

Comments 7: “Why there is no control pregnant women are not included in this study?”

Response 7: Thank you for pointing this out. We acknowledge this limitation. The study aimed specifically to compare PAS and non-invasive placenta previa groups to investigate whether certain early markers could differentiate between varying degrees of placental invasion. Including a control group of normal pregnancies was beyond the scope of this study but is an important consideration for future research. This limitation has now been clearly mentioned in the Discussion section. Additionally, an additional table (Table 4) comparing the findings in our study with the values reported in normal pregnancy in the literature has been added to the Results section.

(See Results page 7, lines 202–204 and Discussion page 7, lines 295-304)

Comments 8: “Is there any specific reason for large differences in the sample size for PAS and PP?”

Response 8: Thank you for pointing this out. Yes, as PAS is a rare condition, the number of PAS cases was naturally much lower than non-invasive PP cases. We have now explained in the Methods section that all eligible PAS cases during the study period were included, resulting in 15 PAS cases versus 166 PP cases without invasion. This natural disparity reflects real-world incidence.

(See Materials and Methods page 3, lines 98–-111)

Comments 9: “How the sample size was determined (based on power analysis or effect size). It should be explained in the sample collection section.”

Response 9: Thank you for pointing this out. We agree that this explanation is important. Due to the retrospective nature of the study and the low incidence of PAS, no a priori power analysis was conducted. Instead, all confirmed cases of PAS within the study period were included. The sample size, although limited, represents the complete eligible population within the institution's database. A statement has been added in the Participants subsection of the Materials and Methods:

“Due to the retrospective nature of the study and the rarity of PAS, no formal a priori power analysis was conducted. Instead, all eligible and confirmed PAS cases during the study period were included. We acknowledge that the relatively small number of PAS cases may limit the statistical power, especially for detecting small-to-moderate differences in secondary variables such as fetal sex distribution. The unequal sample sizes between groups reflect the true incidence of PAS in this population.”

This limitation has also been acknowledged in the Discussion.

(See Materials and Methods page 3, lines 119–124 and Discussion page 9, lines 295–305)

Comments 10: “There is a scope to include a graph or histogram in this manuscript.”

Response 10: Thank you for pointing this out. We have now included a boxplot graph comparing key markers (PAPP-A, β-hCG, MPV) across the PAS and PP groups. This visual aid has been added to the Results section to support interpretation and enhance presentation.

(See Results page 7, lines 207–213)

Comments 11: “Among various hematological parameters, there is a significant change observed only in the MPV parameter? Give justification.”

Response 11: Thank you for pointing this out. Yes, this observation is correct. We have addressed this in the Discussion section. MPV is more sensitive to subtle changes in platelet activation and turnover associated with inflammation and abnormal vascularization. Other parameters like platelet count, NLR, and PLR may be less specific or influenced by multiple confounding factors in pregnancy. This biological rationale has now been briefly discussed.

(See Discussion page 8, lines 273–275)

Comments 12: “Give a brief note on future research directions and how they could address the present study’s limitations in the conclusion.”

Response 12: Thank you for pointing this out. We have added a paragraph in the Conclusion section addressing future directions:

“Larger, prospective multicenter studies with inclusion of normal pregnancy controls are warranted to validate these findings. The integration of biochemical markers with sonographic features, maternal history, and novel proteomic or angiogenic biomarkers could enhance early diagnostic accuracy for PAS. Additionally, exploring machine learning models incorporating these variables may offer promising decision-support tools in prenatal care.”

(See Conclusion page 9, lines 320–325)

Comments 13: “The ethical committee approval was given in the year 2020, but the article is communicated in 2025, is there any specific reason for this delay?”

Response 13: Thank you for pointing this out. The delay between ethical approval and submission occurred due to administrative delays, including data collection, validation, and personnel changes during the COVID-19 pandemic period. These factors extended the timeline for manuscript preparation and submission. A brief explanatory note has been added in the Institutional Review Board Statement section.

Response to Comments on the Quality of English Language

Point 1: (x) The English could be improved to more clearly express the research.

Response 1: We thank the reviewer for this observation. The manuscript has now been professionally edited for English language and grammar to ensure clarity and readability.

Additional clarifications

We thank the reviewer for their thorough and constructive comments, which have helped improve the quality and clarity of our manuscript. We have addressed each point and revised the manuscript accordingly.

Round 2

Reviewer 1 Report

Comments and Suggestions for Authors

No more comments

Reviewer 2 Report

Comments and Suggestions for Authors

The authors answered all the questions in a reasoned manner and made the necessary adjustments to the article. The work is recommended for publication.